# DenseViG: Decoupled Energy-guided Graph Structure Refinement for Vision GNNs

## Abstract

Vision Graph Neural Networks (ViG) treats an image as a set of visual patches for graph representation learning and yields promising results across various computer vision tasks. However, most existing works primarily focus on static graph construction, ignoring the performance gains and noise reduction benefits of dynamic structure refinement. Meanwhile, generative models such as Energy-based Models (EBMs) are generally unsuitable for discriminative tasks and struggle with large-scale images. Our goal is to introduce a unified generative-discriminative paradigm for dynamically modeling relationships between visual patches, aiming to produce higher-quality representations for improving downstream tasks. Specifically, we propose **D**ecoupled **E**nergy **L**earning (DEL) that defines a joint distribution of sample pairs to approximate the target distribution. It decouples EBMs into energy matching and contrastive learning as a global loss function, which pulls similar pairs closer and pushes dissimilar pairs further apart in the representation space. For implementation, we develop an end-to-end framework, termed **D**ecoupled **EN**ergy learning guided **S**tructure r**E**finement for improving **ViG** (DenseViG). Structure refinement is deployed within ViG architectures in a plug-and-play manner, dynamically adding or pruning edges based on similarity metrics with a relaxation strategy. Theoretical analyses demonstrate the effectiveness of DenseViG in processing large datasets through graph operations. Empirical evaluations confirm that it outperforms state-of-the-art methods on three major benchmarks, achieving 84.3% Top-1 accuracy on ImageNet-1K, 46.4% mAP on MS COCO, and 50.9% mIoU on ADE20K.

## Introduction

Graph Neural Networks (GNNs) are initially developed to handle graph-structured data due to their strength in modeling relational dependencies and capturing complex topology. More recently, GNNs have been successfully applied to computer vision Tian et al. (2024); Wang et al. (2023); Rahman & Marculescu (2024); Xu et al. (2024); Bhowmik et al. (2023). Among these efforts, Han et al. Han et al. (2022) pioneered Vision GNNs (ViG), which divides images into patches, constructs a graph over visual features, and propagates information via a message-passing scheme. Rather than relying on traditional 2D convolutions or self-attention mechanisms, ViG adopts graph convolution as a novel paradigm for visual representation learning. It achieves competitive performance across various benchmarks, including image classification Wang et al. (2025); Parikh et al. (2025); Li et al. (2023), object detection Yang et al. (2025); Wu et al. (2023), and image segmentation Li et al. (2024); Jiang et al. (2023). However, the main drawback of ViG lies in its largely static graph construction using the conventional $k$-Nearest-Neighbor ($k$-NN) algorithm and its variants Han et al. (2022); Wang et al. (2025); Yang et al. (2025); Spadaro et al. (2025); Yao et al. (2024). These methods always suffer from fixed topologies, noise interference and limited high-order semantics. To avoid this, Munir et al. (2024) designed a dynamic axial graph construction that connects patches whose Euclidean distance falls below a threshold set by the distance distribution's mean and standard deviation. Han et al. (2023) devised a dynamic hypergraph construction that softly clusters semantically akin patches with Fuzzy C-Means, treating each cluster as a set of hyperedges. Both methods rely on heuristic rules and shallow features without learnable parameters, preventing end-to-end training via gradient updates and restricting the flexibility of ViG.

**Motivation** We aim to comprehensively enhance the performance of ViG on downstream tasks. Prior works typically begin by selecting expressive generative models, particularly Energy-based Models (EBMs) LeCun et al. (2006), known for their strong representational capacity. To better align EBMs with discriminative tasks such as classification and detection Bhowmik et al. (2023); Grathwohl et al. (2019), recent studies by Kim & Ye (2022) and Wang et al. (2022) have explored several combined approaches using contrastive objectives. They have demonstrated effective representations and favorable outcomes on toy datasets (e.g., MNIST and CIFAR). However, its scalability to large-scale datasets is hindered by the time-consuming iterations and high memory consumption associated with EBMs. Inspired by the success of Zeng et al. (2025) in applying this investigation to graph-based domains, we utilize the ViG architecture of converting pixel-level images into sparse graphs.

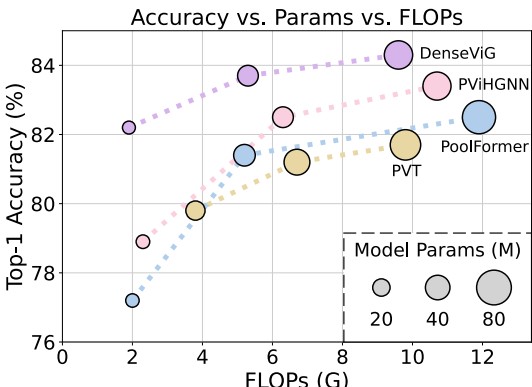

Figure 1: Params, FLOPs, and Top-1 accuracy on ImageNet-1K for representative ViT, ViG, Pool, and HGNN backbones. DenseViG-Ti, S, and M lie on the accuracy and efficiency frontier, reaching 82.2%, 83.7%, and 84.3% Top-1 accuracy. This suggests that decoupled energy-guided graph structure refinement scales to image datasets and points to stronger Vision GNNs.

Since reducing the data dimensionality by several folds, it shortens the iteration time and lowers memory, both of which are beneficial for processing high-resolution imagery. In summary, our perspective positions graph learning as a critical bridge between EBMs and large-scale visual data.

**Contribution** In this paper, we present a Decoupled Energy Learning (DEL) method that addresses the limitations of single-paradigm modeling. DEL incorporates EBMs with Contrastive Learning (CL) to define a joint distribution over paired samples for approximating the target data distribution. Specifically, we decouple its objective as a joint loss including two complementary terms, leveraging the benefits of generative models while retaining strong discriminability for downstream tasks. During energy matching process, DEL objective maximizes the joint log-likelihood of the similarity between positive pairs using EBMs, while simultaneously minimizing that of negative pairs using CL. We theoretically justify that the standard contrastive loss emerges as a special case of the DEL loss when the generative term is omitted.

Building on our motivation, we introduce a novel framework that **D**ecoupled **EN**ergy learning guided **S**tructure r**E**finement for improving **ViG**, named DenseViG. This framework performs graph structure refinement in a modular, plug-and-play fashion, enabling seamless integration into existing ViG architectures without modifying any components. Within each ViG stage, an intrinsic GNN encoder embeds the constructed graph to generate node representations. Dynamic refinement enhances the raw structure by adding or pruning edges according to learned pairwise representation similarities with a Gumbel-Sigmoid relaxation strategy. DEL loss provides a global optimization to facilitate both high-quality representations and optimal structures. DenseViG is evaluated on three benchmarks for image classification, object detection and semantic segmentation. The major contributions are threefold as follows:

**1.** Incorporating EBMs with contrastive learning into DEL approximates the target distribution by pulling similar representations closer and pushing dissimilar ones apart. DEL objective is decoupled into two complementary terms, allowing the unified paradigm to fit more naturally within discriminative tasks while preserving generative capacity.

**2.** By integrating dynamic refinement into each ViG stage in a plug-and-play manner, we propose a novel end-to-end trainable framework, called DenseViG. Our framework encourages ViG architectures to generate powerful representations and refined structures, consistently improving the performance of downstream vision tasks.

**3.** Leveraging graph processing, our framework shortens iteration time and reduces memory, enabling training on large-scale images and achieving superior results across three vision benchmarks. Specifically, DenseViG yields 84.3% Top-1 accuracy on ImageNet-1K, 46.4% mAP on MS COCO, and 50.9% mIoU on ADE20K.

## RELATED WORKS

### VISUAL REPRESENTATION LEARNING

Generative models such as EBMs have shown unique potential in computer vision. EBMs represent the data distribution as a scalar-valued energy function Arbel et al. (2020). Recent advances involve deep neural parameterizations of the energy function, widely applied for image reconstruction Pang et al. (2020), object detection Gustafsson et al. (2021), and human motion modeling Zhang et al. (2025). Optimization of EBMs is often considered challenging, and contrastive techniques have proven effective Gutmann & Hyvärinen (2010). Kim & Ye (2022) introduce a hybrid framework that enriches the SimCLR objective with a generative energy-based term, explicitly modeling the joint distribution of positive pairs rather than relying solely on discriminative alignment. Similarly, Wang et al. (2022) propose a unified probabilistic framework that reinterprets adversarial training as the maximum-likelihood estimation of EBMs, clarifying the generative capability of robust classifiers and bridging supervised adversarial training with contrastive learning. These studies motivate a self-supervised framework that leverages both generative expressiveness and discriminative precision for visual representation learning on small- and medium-scale datasets. However, they still suffer from high computational and memory costs on large-scale data.

### VISION GRAPH NEURAL NETWORKS

Vision graph neural networks have become a rapidly growing alternative to grid-based architectures, allowing flexible interactions between local regions and efficient global context aggregation in images. Among ViG components, graph construction is critical a link of improvement. Static construction methods develop several $k$-NN variants to reduce time and space complexity. WiGNet Spadaro et al. (2025) mitigates the quadratic overhead of vanilla $k$-NN through window partitioning, achieving nearly linear scalability with image size. ClusterViG Parikh et al. (2025) applies the k-means algorithm to cluster image tokens, effectively compressing the search space of $k$-NN. Dynamic graph approaches, including MobileViG Munir et al. (2023) and GreedyViG Munir et al. (2024), replace expensive fully-connected graphs with sparse, structured patch interactions to accelerate computation. These structures dynamically update during the forward passes, improving adaptability across scenarios. More recently, hypergraph methods, such as DVHGNN Li et al. (2025), HgVT Fixelle (2025), and HGFormer Wang et al. (2025), transform patch tokens into hypergraph structures, encoding higher-order relationships through dynamic hypergraph convolutions or attention mechanisms, thus capturing richer semantic information. Unlike the aforementioned approaches, we emphasize structure refinement by utilizing a unified paradigm and integrate it into the existing ViG backbones as an end-to-end framework, enhancing downstream tasks.

## METHODS

In this section, we detail the DenseViG framework. First, we formalize the graph notation and review the concepts of EBMs. We then present the core DEL objective, followed by the structure refinement module and the end-to-end optimization pipeline. Finally, we analyze computational complexity and introduce the three DenseViG variants.

### PRELIMINARIES

**Notations** Given an image $\mathbf{I} \in \mathbb{R}^{H \times W \times 3}$, we partition it into $N$ patches. Each patch is projected into a $D$-dimensional feature $\mathbf{x}_i \in \mathbb{R}^D$ via a learnable linear transformation. Consider a graph $\mathcal{G} = (\mathcal{V}, \mathcal{E}, \mathbf{X})$, where $\mathcal{V} = \{v_1, v_2, \ldots, v_N\}$ is the set of $N = |\mathcal{V}|$ nodes, $\mathcal{E} \subseteq \mathcal{V} \times \mathcal{V}$ is the set of edges, and $\mathbf{X} = \{\mathbf{x}_1, \mathbf{x}_2, \ldots, \mathbf{x}_N\} \in \mathbb{R}^{N \times D}$ contains the set of node features corresponding to each node $v_i$. The adjacency matrix $\mathbf{A} \in \{0, 1\}^{N \times N}$ indicates edge connectivity, where $\mathbf{A}_{ij} = 1$ if nodes $v_i$ and $v_j$ are connected, and $\mathbf{A}_{ij} = 0$ otherwise.

Graph structure learning seeks to jointly generate a refined adjacency matrix $\mathbf{A}^*$ and corresponding representation $\mathbf{H}^*$ for downstream tasks. Notably, this process solely optimizes the raw structure, while keeping all others unchanged.

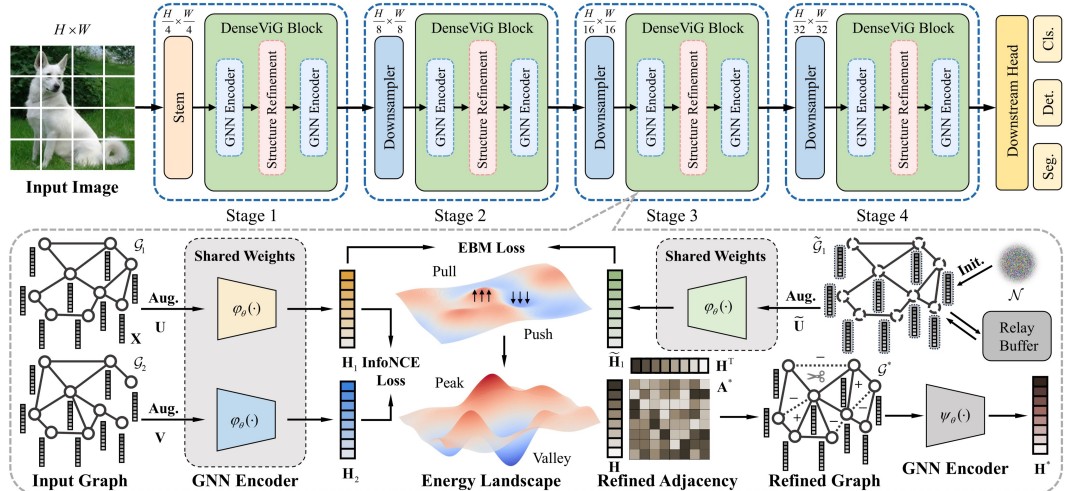

Figure 2: Illustration of the procedure of DenseViG. An image is transformed into node features. Dual graph views feed a shared GNN encoder. DEL improves representations with decoupling EBMs into generative and discriminative aims. Similarities metric and soft relaxation refine structure. Optimal graph boosting downstream task performance in a plug-and-play manner.

**Energy-based Models** define a probability distribution implicitly by assigning a scalar energy to each data point $\boldsymbol{x}$. A parameterized energy function $E_\theta(\boldsymbol{x}) \in \mathbb{R}$ specifies an unnormalized density using the Boltzmann formulation:

$$p_\theta(\boldsymbol{x}) = \frac{\exp(-E_\theta(\boldsymbol{x}))}{Z(\theta)}, \tag{1}$$

where $Z(\theta) = \int \exp(-E_\theta(\boldsymbol{x})) \, \mathrm{d}\boldsymbol{x}$ is the partition function required for normalization.

Since $Z(\theta)$ is usually intractable, training proceeds by calculating the negative log-likelihood of the observed data: $\mathcal{L}(\theta) = \mathbb{E}_{\boldsymbol{x} \sim p_{\text{data}}}[-\log p_\theta(\boldsymbol{x})]$. A popular optimization method is to minimize Kullback–Leibler (KL) divergence Hinton (2002) via gradient descent: $\mathcal{D}_{\text{KL}}(p_{\text{data}} \| p_\theta) = \mathbb{E}_{\boldsymbol{x} \sim p_{\text{data}}}[E_\theta(\boldsymbol{x}) + \log Z(\theta)]$, and its objective becomes:

$$\nabla_\theta \mathcal{D}_{\text{KL}}(p_{\text{data}} \| p_\theta) = \mathbb{E}_{\boldsymbol{x} \sim p_{\text{data}}}[\nabla_\theta E_\theta(\boldsymbol{x})] - \mathbb{E}_{\tilde{\boldsymbol{x}} \sim p_\theta}[\nabla_\theta E_\theta(\tilde{\boldsymbol{x}})]. \tag{2}$$

where true samples $\boldsymbol{x}$ map to lower energies, while generated ones $\tilde{\boldsymbol{x}}$ map to higher energies.

These artificial samples are typically generated by Markov Chain Monte Carlo (MCMC) methods, most often using Langevin dynamics Welling & Teh (2011):

$$\tilde{\boldsymbol{x}}^{k+1} = \tilde{\boldsymbol{x}}^k - \lambda \nabla_{\tilde{\boldsymbol{x}}} E_\theta(\tilde{\boldsymbol{x}}^k) + \xi^k, \tag{3}$$

where $\lambda$ is the step size, and $\xi^k$ is i.i.d. standard Gaussian noise $\mathcal{N}(0, 2\lambda)$. With an appropriate step size $\lambda$ and iterations $k$, the chain converges to the target distribution $p_\theta$.

DECOUPLED ENERGY LEARNING

Let $p_d$ denote the empirical joint data distribution over paired graphs $(\mathcal{G}_1, \mathcal{G}_2)$. To generate two correlated graph views $\mathbf{U}, \mathbf{V} = t_1(\mathbf{X}_1, \mathbf{A}_1), t_2(\mathbf{X}_2, \mathbf{A}_2)$, we independently sample augmentation operators $t_1, t_2 \overset{\text{u.a.r.}}{\sim} \mathcal{T}$, where u.a.r. stands for *uniformly at random*. The augmentation set $\mathcal{T}$ consists of random node feature masking and edge perturbation with a fixed ratio Thakoor et al. (2021).

At each ViG stage, a shared-weight GNN encoder $\varphi_\theta$ takes $\mathbf{U}$ and $\mathbf{V}$ as input and returns node representations $\mathbf{H}_1, \mathbf{H}_2 = \varphi_\theta(\mathbf{U}), \varphi_\theta(\mathbf{V}) \in \mathbb{R}^{N \times F}$.

**Definition.** To approximate $p_d$, a joint distribution $p_\theta$ over the two views $(\mathbf{U}, \mathbf{V})$ can be defined as an EBM:

$$p_\theta(\mathbf{U}, \mathbf{V}) = \frac{\exp(-E_\theta(\mathbf{U}, \mathbf{V}))}{\int \int \exp(-E_\theta(\mathbf{U}, \mathbf{V})) \, \mathrm{d}\mathbf{U} \, \mathrm{d}\mathbf{V}}, \tag{4}$$

where $E_\theta(\mathbf{U}, \mathbf{V}) = \|\mathbf{H}_1 - \mathbf{H}_2\|_2^2 / \tau$ is represented as a metric function that quantifies cross-view similarity, and $\tau$ is a scaling factor.

Concretely, $\|\mathbf{H}_1 - \mathbf{H}_2\|_2^2$ indicates the squared $\ell_2$ distance between the unit-normalized vectors $\mathbf{H}_1$ and $\mathbf{H}_2$. Our core intuition is to minimize this distance for semantically similar pairs from the same sample, while maximizing it for dissimilar pairs from distinct samples. Equivalently, $p_\theta$ assigns higher probability mass to matched pairs under $p_d$ and lower mass to mismatched ones.

The joint negative log-likelihood of Eq. 4 is given by: $\mathbb{E}_{p_d}[-\log p_\theta(\mathbf{U}, \mathbf{V})]$. According to Eq. 2, its corresponding gradient can be written as:

$$\nabla_\theta \mathbb{E}_{p_d}[-\log p_\theta(\mathbf{U}, \mathbf{V})] = \mathbb{E}_{p_d}[\nabla_\theta E_\theta(\mathbf{U}, \mathbf{V})] - \mathbb{E}_{p_\theta}[\nabla_\theta E_\theta(\mathbf{U}, \mathbf{V})] . \tag{5}$$

Eq. 5 requires sampling from $p_\theta(\mathbf{U}, \mathbf{V})$, which is doubly intractable owing to high-dimensional integrals. Naively applying MCMC is prohibitively expensive for the joint target. To avert this issue, we employ the product rule to decouple:

$$\nabla_\theta \mathbb{E}_{p_d}[-\log p_\theta(\mathbf{U}, \mathbf{V})] = \nabla_\theta \mathbb{E}_{p_d}[-\log p_\theta(\mathbf{V}|\mathbf{U})] + \nabla_\theta \mathbb{E}_{p_d}[-\log p_\theta(\mathbf{U})] , \tag{6}$$

where $p_\theta(\mathbf{U})$ denotes the marginal of $p_\theta(\mathbf{U}, \mathbf{V})$.

**Theorem.** The marginal distribution $p_\theta(\mathbf{U})$ is also considered as an EBM:

$$p_\theta(\mathbf{U}) = \frac{\exp(-E_\theta(\mathbf{U}))}{\int \exp(-E_\theta(\mathbf{U})) \, \mathrm{d}\mathbf{U}} , \tag{7}$$

where $E_\theta(\mathbf{U}) = -\log \int \exp(-E_\theta(\mathbf{U}, \mathbf{V})) \, \mathrm{d}\mathbf{V}$.

The negative log-likelihood gradient of Eq. 7 is:

$$\nabla_\theta \mathbb{E}_{p_d}[-\log p_\theta(\mathbf{U})] = \mathbb{E}_{p_d}[\nabla_\theta E_\theta(\mathbf{U})] - \mathbb{E}_{p_\theta}[\nabla_\theta E_\theta(\mathbf{U})] . \tag{8}$$

Following Eq. 6 and Eq. 8, we obtain the gradient of Decoupled Energy Learning (DEL) objective $\mathcal{L}_E(\theta)$, incorporating both the discriminative and generative terms:

$$\nabla_\theta \mathcal{L}_E(\theta) = \nabla_\theta \mathbb{E}_{p_d}[-\log p_\theta(\mathbf{V}|\mathbf{U})] + \alpha[\mathbb{E}_{p_d}[\nabla_\theta E_\theta(\mathbf{U})] - \mathbb{E}_{p_\theta}[\nabla_\theta E_\theta(\mathbf{U})]] , \tag{9}$$

where $\alpha$ is a hyper-parameter to balance the two terms.

Eq. 9 allows DEL to indirectly solve the intractable joint distribution. The discriminative term is straightforward without computing the global gradients w.r.t. $Z(\theta)$, and the generative term only requires marginal sampling over single view. Negative samples are synthesized via Langevin dynamics from Gaussian noise initialization, as shown in Eq. 3. Besides, discriminative alignment alone may eliminate the energy gap between positive and negative samples, resulting in representation collapse. Generative regularization encourages higher energy for mismatched pairs using contrastive divergence. This push-pull mechanism accurately aligns the energy landscape with the underlying data manifold. More proof details are provided in the Appendix A.

STRUCTURE REFINEMENT

Given node representations $\mathbf{H} = \{\mathbf{h}_1, \mathbf{h}_2, \ldots, \mathbf{h}_N\}$, we calculate the cosine-similarity matrix $\mathbf{C}_{ij} = \frac{\mathbf{h}_i \mathbf{h}_j^\top}{\|\mathbf{h}_i\|_2 \|\mathbf{h}_j\|_2}$. To obtain a differentiable adjacency matrix, we adopt a Relaxed Bernoulli variable for each edge via the Gumbel–Sigmoid reparameterization Jang et al. (2016):

$$\mathbf{S} = \mathrm{Sigmoid}\left(\frac{1}{\pi}\left(\log \frac{\mathbf{C}}{1 - \mathbf{C}} + \log \frac{\delta}{1 - \delta}\right)\right) , \tag{10}$$

where $\delta \sim \mathrm{Uniform}(0, 1)$, and $\pi \in \mathbb{R}^+$ controls the relaxation sharpness. As $\pi \to 0$, $\mathbf{S}$ converges to a hard binary decision. For structural validity and numerical stability, we symmetrize and normalize $\mathbf{S}$: $\mathbf{A}^* = \mathbf{D}^{-\frac{1}{2}}\left(\frac{\mathbf{S} + \mathbf{S}^\top}{2}\right)\mathbf{D}^{-\frac{1}{2}}$, where $\mathbf{D}$ is the diagonal degree matrix. The refined graph $\mathcal{G}^* = (\mathbf{X}, \mathbf{A}^*)$ is passed to a GNN encoder $\psi_\theta$, architecturally identical to $\varphi_\theta$, to yield the representation $\mathbf{H}^*$ for the subsequent processing.

END-TO-END OPTIMIZATION

For the implementation of DEL, we design a tractable MCMC estimator in which the generative term is rewritten as contrastive form, and the discriminative term is expressed as an InfoNCE loss.

Given a mini-batch of $N$ paired augmentations $\{\mathbf{U}_i, \mathbf{V}_i\}_{i=1}^N \sim p_d$, each $(\mathbf{U}_i, \mathbf{V}_i)$ forms a positive pair, while $(\mathbf{U}_i, \mathbf{V}_j)$ serve as negative ones for the discriminative term. To balance the two phases of the generative term in KL divergence, we synthesize $N$ views $\{\tilde{\mathbf{U}}_i\}_{i=1}^N \sim p_\theta$ using $k$ Langevin dynamics steps in the negative phase.

Since the marginal energy $E_\theta(\mathbf{U}) = -\log \int \exp(-E_\theta(\mathbf{U}, \mathbf{V})) \, \mathrm{d}\mathbf{V}$ is intractable, we approximate it with a log-sum-exp over $M$ complementary views $\{\mathbf{V}_{i,m}\}_{m=1}^M$ from the same batch: $\hat{E}_\theta(\mathbf{U}_i) \approx -\log \frac{1}{M} \sum_{m=1}^M \exp(-E_\theta(\mathbf{U}_i, \mathbf{V}_{i,m}))$. Substituting this into Eq. 9 gives the generative loss:

$$\mathcal{L}_{\text{Gen}}(\theta) \approx \frac{1}{N} \sum_{i=1}^N (\hat{E}_\theta(\mathbf{U}_i) - \hat{E}_\theta(\tilde{\mathbf{U}}_i)) . \tag{11}$$

Treating each positive pair $(\mathbf{U}_i, \mathbf{V}_i^+)$ against $M'$ intra-batch negatives $\{\mathbf{V}_{i,j}^-\}_{j=1}^{M'}$ yields the discriminative loss:

$$\mathcal{L}_{\text{Disc}}(\theta) \approx -\frac{1}{N} \sum_{i=1}^N \log \frac{\exp(-E_\theta(\mathbf{U}_i, \mathbf{V}_i^+))}{\exp(-E_\theta(\mathbf{U}_i, \mathbf{V}_i^+)) + \sum_{j=1}^{M'} \exp(-E_\theta(\mathbf{U}_i, \mathbf{V}_{i,j}^-))} . \tag{12}$$

In summary, the complete DEL objective is: $\mathcal{L}_E(\theta) = \mathcal{L}_{\text{Disc}}(\theta) + \alpha \mathcal{L}_{\text{Gen}}(\theta) + \beta \mathcal{L}_r(\theta)$, where $\mathcal{L}_r$ is the $\ell_2$ regularization that prevents gradient overflow from large energy values. For end-to-end training, we minimize the global loss function: $\mathcal{L}_G = \mathcal{L}_T + \mu \mathcal{L}_E + \gamma \mathcal{L}_{\mathcal{R}}$, where $\mathcal{L}_T$ is the task-specific loss, $\mathcal{L}_{\mathcal{R}} = \sum_{i,j} \mathbb{E}_\delta[\mathbf{S}_{ij}]$ is a sparsity-promoting penalty term, $\mu$ and $\gamma$ are hyperparameters. The pseudocode of DenseViG is illustrated in Algorithm 1 in the Appendix C.1.

ARCHITECTURE OVERVIEW

As illustrated in Fig. 2, DenseViG is a four-stage architecture that enriches a PyramidViG backbone Han et al. (2022) with DEL-guided structure refinement. We introduce the model's scaling variants and provide a brief complexity analysis. The complete configuration is in the Appendix C.2.

**Scaling Variants** Mirroring the design philosophy of ViG, we instantiate three different configurations: DenseViG-Ti, DenseViG-S, and DenseViG-M. Across all variants, the initial node number is fixed to $N = 3136$, while later stages reduce it via spatial downsampling. We set 9 nearest neighbors in the $k$-NN graph construction, and employ 4 attention heads. Table 1 summarizes the depth, blocks per stage, and dimensionality of each model.

Table 1: Scaling variants of DenseViG. $d_D$ and $d_F$ is the feature and representation dimensions, respectively. 'Ti', 'S', and 'M' stand for Tiny, Small, and Medium.

| Model | Depth | Blocks | Dim. $d_D$ | Dim. $d_F$ |
|---|---|---|---|---|
| DenseViG-Ti | 12 | [2, 2, 6, 2] | 192 | [48, 96, 240, 384] |
| DenseViG-S | 12 | [2, 2, 6, 2] | 320 | [80, 160, 400, 640] |
| DenseViG-M | 22 | [2, 2, 16, 2] | 640 | [96, 192, 384, 768] |

**Complexity Analysis** DenseViG introduces additional computation in only two modules: structure refinement computes $B$ cosine-similarity matrices of size $N \times N$, with a cost of $\mathcal{O}(B \cdot N^2 \cdot F)$, where $F$ is the representation dimension. Since ViG already condenses an $H \times W$ image into $N$ nodes, this quadratic term is empirically affordable. DEL performs $k$ Langevin steps on the $B$ graph views in a mini-batch, adding a linear overhead of $\mathcal{O}(k \cdot B \cdot N \cdot F)$. With a small $k$ and $F$, this cost is negligible relative to structure refinement. Therefore, the total overhead is $\mathcal{O}(B \cdot N^2 \cdot F + k \cdot B \cdot N \cdot F)$, and the overall pipeline retains near-linear scalability w.r.t. image size.

EXPERIMENTS

IMAGE CLASSIFICATION

**Implementation Details** All DenseViG variants are trained on ImageNet-1K Deng et al. (2009), which comprises 1.2M training images, 50K validation images, and 100K test images across 1K

Table 2: Image classification performance on ImageNet-1K. ♠ CNN, ♦ ViT, ♥ Pool, ♣ GNN, ❌ CNN-GNN, ▼ HGNN, and ★ DenseViG (ours). Reporting Params (M), FLOPs (G), and Top-1 & Top-5 Accuracy (%) where available.

| Model | Params (M) | FLOPs (G) | Top-1 | Top-5 | Model | Params (M) | FLOPs (G) | Top-1 | Top-5 |
|---|---|---|---|---|---|---|---|---|---|
| ♠ ResNet18 | 11.7 | 1.8 | 70.6 | 89.7 | ♥ PoolFormer-S12 | 12.0 | 2.0 | 77.2 | 93.5 |
| ♠ ResNet50 | 25.6 | 4.1 | 79.8 | 95.0 | ♥ PoolFormer-S36 | 31.0 | 5.2 | 81.4 | 95.5 |
| ♠ ResNet152 | 60.2 | 11.5 | 81.8 | 95.9 | ♥ PoolFormer-M48 | 73.0 | 11.9 | 82.5 | 96.0 |
| ♠ ConvNeXt-T | 29.0 | 4.5 | 82.1 | 95.9 | ❌ MobileViG-Ti | 5.2 | 0.7 | 75.7 | - |
| ♠ ConvNeXt-S | 50.0 | 8.7 | 83.1 | 96.4 | ❌ MobileViG-S | 7.2 | 1.0 | 78.2 | - |
| ♠ ConvNeXt-B | 89.0 | 15.4 | 83.8 | 96.7 | ❌ MobileViG-M | 14.0 | 1.5 | 80.6 | - |
| ♦ PVT-Ti | 13.2 | 1.9 | 75.1 | - | ❌ GreedyViG-S | 12.0 | 1.6 | 81.1 | - |
| ♦ PVT-S | 24.5 | 3.8 | 79.8 | - | ❌ GreedyViG-M | 21.9 | 3.2 | 82.9 | - |
| ♦ PVT-M | 44.2 | 6.7 | 81.2 | - | ❌ GreedyViG-B | 30.9 | 5.2 | 83.9 | - |
| ♦ DeiT-Ti | 5.7 | 1.3 | 72.2 | 80.1 | ▼ PViHGNN-Ti | 12.3 | 2.3 | 78.9 | 94.6 |
| ♦ DeiT-S | 22.1 | 4.6 | 79.8 | 85.7 | ▼ PViHGNN-S | 28.5 | 6.3 | 82.5 | 96.3 |
| ♦ DeiT-B | 86.4 | 17.6 | 81.8 | 86.7 | ▼ PViHGNN-M | 52.4 | 10.7 | 83.4 | 96.5 |
| ♦ Swin-T | 29.0 | 4.5 | 81.3 | 95.5 | ▼ DVHGNN-T | 11.1 | 1.9 | 79.8 | - |
| ♦ Swin-S | 50.0 | 8.7 | 83.0 | 96.2 | ▼ DVHGNN-S | 30.2 | 5.2 | 83.1 | - |
| ♦ Swin-B | 88.0 | 15.4 | 83.5 | 96.5 | ▼ DVHGNN-M | 52.5 | 10.4 | 83.8 | - |
| ♣ PyramidViG-Ti | 10.7 | 1.7 | 78.2 | 94.2 | ▼ HgVT-Ti | 7.7 | 1.8 | 76.2 | 93.2 |
| ♣ PyramidViG-S | 27.3 | 4.6 | 82.1 | 96.0 | ▼ HgVT-S | 22.9 | 5.5 | 81.2 | 95.5 |
| ♣ PyramidViG-M | 51.7 | 8.9 | 83.1 | 96.4 | ★ DenseViG-Ti | 11.1 | 1.9 | **82.2** | **96.1** |
| ♣ WiGNet-Ti | 10.8 | 2.1 | 78.8 | 94.6 | ★ DenseViG-S | 28.5 | 5.3 | **83.7** | **96.7** |
| ♣ WiGNet-S | 27.4 | 5.7 | 82.0 | 95.9 | ★ DenseViG-M | 53.3 | 9.6 | **84.3** | **96.9** |
| ♣ WiGNet-M | 49.7 | 11.2 | 83.0 | 96.3 | | | | | |

classes. Training is conducted for 310 epochs on $224 \times 224$ crops with the AdamW optimizer (momentum $\beta_1$=0.9, $\beta_2$=0.99, weight decay=0.05). The learning rate is initialized at 0.001 and annealed by a cosine scheduler. We apply RandAugment, CutMix, RandomResizedCrop, and label smoothing for data augmentation, and track an Exponential Moving Average (EMA) of model parameters. All experiments are implemented in PyTorch and run on 8 RTX3090 GPUs with a batch size of 64. The hyperparameters are specified in the Appendix C.3. to ensure the reproducibility.

**Experimental Results** Table 2 compares DenseViG with leading image classification methods, including ResNet He et al. (2016), ConvNeXt Liu et al. (2022d), PVT Wang et al. (2021b), DeiT Touvron et al. (2021), Swin Liu et al. (2022c), PoolFormer Yu et al. (2022), PyramidViG Han et al. (2022), WiGNet Spadaro et al. (2025), MobileViG Munir et al. (2023), GreedyViG Munir et al. (2024), PViHGNN Han et al. (2023), DVHGNN Li et al. (2025), and HgVT Fixelle (2025). Notably, DenseViG-S boosts Top-1 accuracy by 1.7% over the strongest GNN baseline WiGNet-S and by 0.6% over the best HGNN baseline DVHGNN-S. These margins confirm that DenseViG captures higher-order visual information interactions more effectively than prior approaches, without appreciably increasing model size or computational cost.

OBJECT DETECTION

**Implementation Details** Object detection is conducted on MS COCO Lin et al. (2014), containing 118K training, 5K validation, and 20K test images. RetinaNet Lin et al. (2017) and Mask R-CNN He et al. (2017) are employed as detection heads and implemented in the MMDetection Chen et al. (2019). All models are pre-trained on ImageNet-1K and fine-tuned for 12 epochs at the resolution of $1280 \times 800$ with an AdamW optimizer, an initial learning rate of $2 \times 10^{-4}$, and an effective batch size of 16. We additionally apply data augmentations including random horizontal Flipping and Multi-scale jittering. Other settings are the same as the image classification setup. Bounding box regression uses a weighted sum of $\ell_1$ and GIoU losses.

**Experimental Results** As depicted in Table 3, DenseViG outperforms the state-of-the-art approaches. Specifically, under the RetinaNet head, DenseViG-S achieves 44.6% mAP that surpasses PyramidViG by 2.8%, ViHGNN-S by 2.4%, and PViHGNN-S by 1.3%, respectively. Under the Mask R-CNN head, DenseViG-S achieves 46.4% mAP that surpasses PyramidViG by 3.8%, ViHGNN-S by 3.3%, and PViHGNN-S by 1.6%, respectively. More results are in the Appendix D.

Table 3: Object detection performance on MS COCO. 1× denotes the standard training schedule of 12 epochs. Indicating Params (M), FLOPs (G), and mAP (%) where available.

| Backbone | RetinaNet 1× | | | | |
| --- | --- | --- | --- | --- | --- |
| | Params | FLOPs | $AP^{box}$ | $AP^{box}_{50}$ | $AP^{box}_{75}$ |
| ResNet-50 | 38M | 239G | 36.3 | 55.3 | 38.6 |
| PVT-S | 34M | 227G | 40.4 | 61.3 | 44.2 |
| Swin-T | 39M | 245G | 41.5 | 62.1 | 44.2 |
| PyramidViG-S | 36M | 240G | 41.8 | 63.1 | 44.7 |
| PViHGNN-S | 38M | 244G | 42.2 | 63.8 | 45.1 |
| DVHGNN-S | 38M | 242G | 43.3 | 64.3 | 46.3 |
| DenseViG-S | 40M | 247G | **44.6** | **66.1** | **47.8** |

| Backbone | Mask R-CNN 1× | | | | |
| --- | --- | --- | --- | --- | --- |
| ResNet-50 | 44M | 260G | 38.0 | 58.6 | 41.4 |
| PVT-S | 44M | 245G | 40.4 | 62.9 | 43.8 |
| Swin-T | 48M | 264G | 42.2 | 64.6 | 46.2 |
| PyramidViG-S | 46M | 259G | 42.6 | 65.2 | 46.0 |
| PViHGNN-S | 48M | 262G | 43.1 | 66.0 | 46.5 |
| DVHGNN-S | 49M | 261G | 44.8 | 66.8 | 49.0 |
| DenseViG-S | 51M | 266G | **46.4** | **68.7** | **50.6** |

Table 4: Semantic segmentation performance on ADE20k. Head method includes input configuration: -1 last backbone layer only, -4 last four layers concatenated. Showing Params (M), mIoU (%), and Pixel Accuracy (%) where available.

| Backbone | Params | Head | Params | mIoU | Acc. |
| --- | --- | --- | --- | --- | --- |
| ResNet-18 | 11.5M | PPM-1 | 12.9M | 33.8 | 76.1 |
| ResNet-50 | 25.6M | SFPN | 1.6M | 36.7 | - |
| ResNet-50 | 25.6M | PPM-1 | 23.2M | 41.3 | 79.7 |
| ResNet-101 | 44.5M | PPM-1 | 23.2M | 42.2 | 80.6 |
| ConvNeXt-T | 29.0M | UperNet | 29.8M | 46.1 | - |
| PVT-S | 24.5M | SFPN | 1.6M | 39.8 | - |
| PoolFormer-S12 | 12.0M | SFPN | 1.6M | 37.2 | - |
| Swin-T | 28.3M | SFPN | 1.6M | 41.5 | - |
| Swin-T | 28.3M | UperNet | 29.8M | 46.1 | - |
| GreedyViG-S | 12.0M | SFPN | 1.6M | 43.2 | - |
| GreedyViG-B | 30.9M | SFPN | 1.6M | 47.4 | - |
| HgVT-S | 22.9M | MLP-4 | 235K | 28.5 | 71.8 |
| HgVT-S | 22.9M | PPM-4 | 15.5M | 36.0 | 75.7 |
| DenseViG-Ti | 11.1M | MLP-4 | 235K | **33.0** | **76.8** |
| DenseViG-Ti | 11.1M | SFPN | 1.6M | **42.7** | **79.4** |
| DenseViG-S | 28.5M | PPM-4 | 15.5M | **48.8** | **81.7** |
| DenseViG-S | 28.5M | UperNet | 29.8M | **50.9** | **82.5** |

## SEMANTIC SEGMENTATION

**Implementation Details** We validate semantic segmentation on ADE20K Zhou et al. (2017), which provides 20K training, 2K validation, and 3K test images covering 150 categories. The backbone is initialized with ImageNet-1K weights and fine-tuned for 16 epochs with 4 segmentation heads: MLP, PPM Zhao et al. (2017), Semantic FPN Kirillov et al. (2019), and UPerNet Xiao et al. (2018). All models are implemented in the MMSegmentation Contributors (2020). Training uses $512 \times 512$ crops with a batch size of 16, and the same data augmentation pipeline, GPU configuration, optimizer settings, and learning rate schedule as in our object detection setup. Supervision is provided by the standard pixel-wise cross-entropy loss.

**Experimental Results** Table 4 presents the superiority of DenseViG with other famous backbones. Specifically, using lightweight linear heads such as MLP and PPM, DenseViG-Ti achieves competitive results with faster inference speed. With SFPN and UPerNet head, DenseViG-S achieves 48.8% and 50.9% mIoU, outperforming the suboptimal GreedyViG-B by 1.4% and 3.5%, respectively.

## ABLATION STUDIES

**Effectiveness of DEL Objective** Table 5 investigates the influence of generative and discriminative terms in DEL on classification performance. Using only the generative term improves Top-1 accuracy by 0.5-3.1% over the baseline, suggesting EBMs enhance representation quality. Replacing EBMs with InfoNCE independently achieves accuracies of 81.7% and 83.0% on ViG-Ti and ViG-S, demonstrating the effectiveness of the discriminative term. Involving EBMs with SimCLR yields a 0.6% gain, while pairing EBMs

Table 5: Performance ablations of generative and discriminative terms in DEL on ImageNet-1K using the DenseViG-Ti/S backbone. Denoting used (✓) or not used (✗).

| Method | Gen. term | Disc. term | | Top-1 |
| --- | --- | --- | --- | --- |
| | EBMs | InfoNCE | SimCLR | |
| PyramidViG-Ti | ✗ | ✗ | ✗ | 78.2 |
| DenseViG-Ti | ✓ | ✗ | ✗ | 81.3 |
| DenseViG-Ti | ✗ | ✓ | ✗ | 81.7 |
| DenseViG-Ti | ✓ | ✗ | ✓ | **82.4** |
| DenseViG-Ti | ✓ | ✓ | ✗ | 82.2 |
| PyramidViG-S | ✗ | ✗ | ✗ | 82.1 |
| DenseViG-S | ✓ | ✗ | ✗ | 82.6 |
| DenseViG-S | ✗ | ✓ | ✗ | 83.0 |
| DenseViG-S | ✓ | ✗ | ✓ | 83.6 |
| DenseViG-S | ✓ | ✓ | ✗ | **83.7** |

with InfoNCE obtains the optimal balance of 82.2% and 83.7%. These findings reinforce our view that standard contrastive loss represents a special case of DEL.

**Effectiveness of Structure Refinement** Table 6 presents the impact of varying refinement strategy. Static $k$-NN and the axial graph score 82.1% and 81.1%. With $\pi$=0.1, adding edges only lifts accuracy to 83.3%, while pruning edges only reaches 82.9%, proving that each step alone reduces noise or restores missing links. Applying both adding and pruning at every stage pushes accuracy to 83.7%, surpassing the half-stage schedule and the milder relaxation. Full-stage refinement with a softer relaxation therefore yields the best result.

Table 6: Impact of various structure refinement methods on ImageNet-1K. "Half" means placing at every second stage. The temperature $\pi$ controls relaxation sharpness.

| Model | Refinement | Relaxation | Stage | Top-1 |
|---|---|---|---|---|
| PyramidViG-S | Static $k$-NN | Hard | All | 82.1 |
| GreedyViG-S | Axial graph | Hard | All | 81.1 |
| DenseViG-S | Add-only | $\pi$=0.1 | All | 83.3 |
| DenseViG-S | Prune-only | $\pi$=0.1 | All | 82.9 |
| DenseViG-S | Add-prune | $\pi$=0.1 | Half | 83.1 |
| DenseViG-S | Add-prune | $\pi$=1.0 | All | 83.5 |
| DenseViG-S | Add-prune | $\pi$=0.1 | All | **83.7** |

**Analysis of Iterative Steps** Fig. 3 shows that raising $k$ from 1 to 5 lifts Top 1 accuracy by 1.0%, adding 79 ms and 8.22 GB. Increasing to 10 or 20 yields little extra accuracy but much higher cost. We choose $k$=5, which provides near peak accuracy with acceptable time and memory, confirming the scalability of our EBM sampler on current environments.

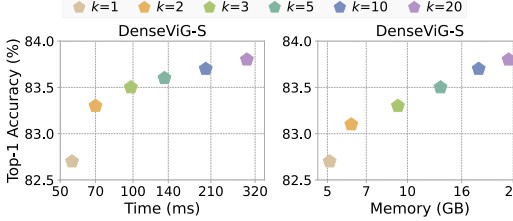 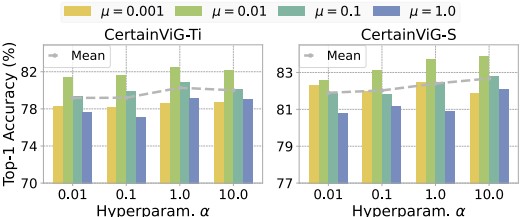

Figure 3: Impact of $k$ Langevin steps in negative sampling for DenseViG-S on ImageNet-1K. Evaluated on 8 RTX 3090 GPUs (batch size 64).

Figure 4: Hyper-parameter $\alpha$ and $\mu$ analysis of DenseViG-Ti and DenseViG-S on ImageNet-1K. 'Mean' reports the average of the four columns.

**Setting of Hyper-parameters** Fig. 4 indicates that raising $\alpha$ from 0.01 to 10.0 significantly improves Top 1 accuracy of DenseViG-Ti and S. Performance peaks at $\alpha$=10.0, yet generated samples reduce diversity, which means an excessive discriminative weight weakens the EBM's generative power. We therefore set $\alpha$=1.0 for a trade-off outcome. Increasing $\mu$ from 0.001 to 0.1 brings rapid gains and confirms the benefit of the Dense framework. Above 0.1 the improvement levels off and training slows, so we fix $\mu$=0.1 for all models.

Table 7: Impact of adding Dense to two ViG backbones on ImageNet-1K, with absolute gains shown in parentheses.

| Backbone | Params | FLOPs | Top-1 |
|---|---|---|---|
| WiGNet-S (baseline) | 27.4M | 5.7G | 82.0 |
| WiGNet-S + **Dense** | 28.6M | 6.4G | **83.5 (+1.5)**↑ |
| GreedyViG-S (baseline) | 12.0M | 1.6G | 81.1 |
| GreedyViG-S + **Dense** | 12.6M | 1.8G | **82.8 (+1.7)**↑ |

**Plug-and-play Verification** Table 7 demonstrates a 1.5% Top-1 gain on WiGNet-S and 1.7% on GreedyViG-S. Added parameters and FLOPs are minimal, confirming that Dense framework can be seamlessly plugged into existing backbones and boosts performance with minimal extra cost.

## CONCLUSION

This paper proposes DenseViG, a novel end-to-end framework that decouples EBMs into generative and discriminative aims to enable differentiable graph structure refinement for enhancing ViG. At each stage, DenseViG dynamically adds or prunes edges between visual patches, yielding cleaner and higher-quality representations while suppressing noise. The resulting sparse graphs significantly decrease the computational and memory costs of EBMs on high-resolution images. Extensive experiments on three vision benchmarks show that DenseViG sets new state-of-the-art results, validating the effectiveness of decoupled energy-guided graph structure refinement.

## ETHICS STATEMENT

All authors have read and agree to follow the ICLR Code of Ethics. This work aims to advance representation learning for the public good. We use only public data and do not process personal data. We respect licenses and uphold fairness, privacy, security, and research integrity.

## REPRODUCIBILITY STATEMENT

We will release the code after acceptance. Our experiments use public data and standard settings. We will share scripts, configuration files, and clear steps so others can reproduce the results. All resources will have transparent sources.

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

APPENDIX

# A  MISSING PROOFS AND DERIVATIONS

## A.1  REFORMULATION OF THE MARGINAL EBM OBJECTIVE AS A GENERATIVE LOSS

Starting from Eq. 1 and the marginal probability $p_\theta(\mathbf{U})$, we have

$$
\begin{aligned}
\nabla_\theta \log p_\theta(\mathbf{U}) &= \nabla_\theta \log \frac{\exp(-E_\theta(\mathbf{U}))}{Z_\theta} \\
&= -\nabla_\theta E_\theta(\mathbf{U}) - \nabla_\theta \log Z_\theta \\
&= -\nabla_\theta E_\theta(\mathbf{U}) - \nabla_\theta \log \sum_{\mathbf{U}} \exp(-E_\theta(\mathbf{U})) \\
&= -\nabla_\theta E_\theta(\mathbf{U}) - \frac{\sum_{\mathbf{U}} \nabla_\theta \exp(-E_\theta(\mathbf{U}))}{\sum_{\mathbf{U}'} \exp(-E_\theta(\mathbf{U}'))} \\
&= -\nabla_\theta E_\theta(\mathbf{U}) + \sum_{\mathbf{U}} \frac{\exp(-E_\theta(\mathbf{U}))}{\sum_{\mathbf{U}'} \exp(-E_\theta(\mathbf{U}'))} \nabla_\theta E_\theta(\mathbf{U}) \\
&= -\nabla_\theta E_\theta(\mathbf{U}) + \sum_{\mathbf{U}} p_\theta(\mathbf{U}) \nabla_\theta E_\theta(\mathbf{U}) \\
&= -\nabla_\theta E_\theta(\mathbf{U}) + \mathbb{E}_{\tilde{\mathbf{U}} \sim p_\theta}[\nabla_\theta E_\theta(\tilde{\mathbf{U}})],
\end{aligned}
$$

where we use $\nabla_\theta \log Z_\theta = -\mathbb{E}_{\tilde{\mathbf{U}} \sim p_\theta}[\nabla_\theta E_\theta(\tilde{\mathbf{U}})]$. Therefore, the gradient of the negative log-likelihood $\mathcal{L}_\theta$ under the data distribution $p_d$ is

$$
\nabla_\theta \mathcal{L}_\theta = -\nabla_\theta \mathbb{E}_{\mathbf{U} \sim p_d}[\log p_\theta(\mathbf{U})] = \mathbb{E}_{\mathbf{U} \sim p_d}[\nabla_\theta E_\theta(\mathbf{U})] - \mathbb{E}_{\tilde{\mathbf{U}} \sim p_\theta}[\nabla_\theta E_\theta(\tilde{\mathbf{U}})].
$$

Integrating both sides shows that $\mathcal{L}_\theta$ is equivalent (up to a $\theta$-independent constant) to an energy matching objective, corresponding to the generative loss $\mathcal{L}_{\mathrm{Gen}}(\theta)$:

$$
\mathcal{L}_{\mathrm{Gen}}(\theta) = \mathbb{E}_{\mathbf{U} \sim p_d}[E_\theta(\mathbf{U})] - \mathbb{E}_{\tilde{\mathbf{U}} \sim p_\theta}[E_\theta(\tilde{\mathbf{U}})] + \mathrm{const} \approx \frac{1}{N} \sum_{i=1}^{N} E_\theta(\mathbf{U}_i) - \frac{1}{M} \sum_{j=1}^{M} E_\theta(\tilde{\mathbf{U}}_j).
$$

Finally, minimizing $\mathcal{L}_{\mathrm{Gen}}(\theta)$ is equivalent to minimizing the KL divergence $\mathcal{D}_{\mathrm{KL}}(p_d \| p_\theta)$, since

$$
\begin{aligned}
\nabla_\theta \mathcal{D}_{\mathrm{KL}}(p_d \| p_\theta) &= \nabla_\theta \sum_{\mathbf{U}} p_d(\mathbf{U}) \log \frac{p_d(\mathbf{U})}{p_\theta(\mathbf{U})} \\
&= \nabla_\theta \sum_{\mathbf{U}} p_d(\mathbf{U}) \log p_d(\mathbf{U}) - \nabla_\theta \sum_{\mathbf{U}} p_d(\mathbf{U}) \log p_\theta(\mathbf{U}) \\
&= \nabla_\theta \mathbb{E}_{\mathbf{U} \sim p_d(\mathbf{U})}[\log p_d(\mathbf{U})] - \nabla_\theta \mathbb{E}_{\mathbf{U} \sim p_d(\mathbf{U})}[\log p_\theta(\mathbf{U})] \\
&= 0 + \nabla_\theta \mathcal{L}_{\mathrm{Gen}}(\theta) \\
&= \nabla_\theta \mathcal{L}_{\mathrm{Gen}}(\theta),
\end{aligned}
$$

where $\mathbb{E}_{\mathbf{U} \sim p_d}[\log p_d(\mathbf{U})]$ is $\theta$-independent.

## A.2  REGULARIZATION FOR THE EBM OBJECTIVE

Following Du & Mordatch (2019), we add an $\ell_2$ penalty on the magnitudes of the energy values for both real and model samples when optimizing $\mathcal{L}_E(\theta)$. While the relative energy differences between true and negative samples shape the energy landscape, the absolute scale of energies can drift during training, causing large values and numerical instability. The penalty anchors the scale and improves gradient behavior in both positive and negative phases. Concretely, real samples $\mathbf{U}_i \sim p_d(\mathbf{U})$ and negatives $\tilde{\mathbf{U}}_j \sim p_\theta(\tilde{\mathbf{U}})$ in the energy function, we use

$$
\mathcal{L}_r(\theta) = \frac{1}{N} \sum_{i=1}^{N} E_\theta(\mathbf{U}_i)^2 + \frac{1}{M} \sum_{j=1}^{M} E_\theta(\tilde{\mathbf{U}}_j)^2,
$$

where $N$ and $M$ are the mini-batch sizes for real and negative samples, respectively. In the overall objective, this term is weighted by $\beta$ to prevent gradient overflow and stabilize Langevin dynamics.

### A.3 Derivation of the Conditional EBM Objective as a Discriminative loss

The discriminative term of $\mathcal{L}_E$ can be defined as the expected negative log-likelihood of the target $\mathbf{V}$ given context $\mathbf{U}$:

$$\mathcal{L}_{\mathrm{Disc}}(\theta) = \mathbb{E}_{(\mathbf{U},\mathbf{V})\sim p_d}[-\log p_\theta(\mathbf{V}|\mathbf{U})].$$

Since the conditional probability $p_\theta(\mathbf{V}|\mathbf{U}) = p_\theta(\mathbf{U},\mathbf{V})/p_\theta(\mathbf{U})$, we can rewrite this as

$$\mathcal{L}_{\mathrm{Disc}}(\theta) = \mathbb{E}_{p_d}[-\log \frac{p_\theta(\mathbf{U},\mathbf{V})}{p_\theta(\mathbf{U})}].$$

Consider the conditional EBM over paired views $(\mathbf{U},\mathbf{V})$:

$$p_\theta(\mathbf{V}|\mathbf{U}) = \frac{\exp(-E_\theta(\mathbf{U},\mathbf{V}))}{Z_\theta(\mathbf{U})}, \qquad Z_\theta(\mathbf{U}) = \int \exp(-E_\theta(\mathbf{U},\mathbf{V}'))\,d\mathbf{V}',$$

we have

$$p_\theta(\mathbf{V}|\mathbf{U}) = \frac{\exp(-E_\theta(\mathbf{U},\mathbf{V}))}{\int \exp(-E_\theta(\mathbf{U},\mathbf{V}'))\,d\mathbf{V}'}.$$

Hence, the discriminative loss can be written as

$$\mathcal{L}_{\mathrm{Disc}}(\theta) = \mathbb{E}_{p_d}[-\log p_\theta(\mathbf{V}|\mathbf{U})] = \mathbb{E}_{p_d}[-\log \frac{\exp(-E_\theta(\mathbf{U},\mathbf{V}))}{\int \exp(-E_\theta(\mathbf{U},\mathbf{V}'))\,d\mathbf{V}'}].$$

In practice, the expectation over $p_d$ is estimated by the empirical average over $N$ data pairs $\{\mathbf{U}_i,\mathbf{V}_i\}_{i=1}^{N}$. Instead of approximating the intractable normalizer, we optimize the InfoNCE surrogate by normalizing over one positive and several negatives. Concretely, for each positive pair $(\mathbf{U}_i,\mathbf{V}_i^+)$, we select $M'$ negative samples $\{\mathbf{V}_{i,j}^-\}_{j=1}^{M'}$:

$$\mathcal{L}_{\mathrm{Disc}}(\theta) \approx -\frac{1}{N}\sum_{i=1}^{N}\log \frac{\exp(-E_\theta(\mathbf{U}_i,\mathbf{V}_i^+))}{\exp(-E_\theta(\mathbf{U}_i,\mathbf{V}_i^+)) + \sum_{j=1}^{M'}\exp(-E_\theta(\mathbf{U}_i,\mathbf{V}_{i,j}^-))},$$

where $\mathbf{V}_i^+$ denotes the positive response for $\mathbf{U}_i$, and $\mathbf{V}_{i,j}^-$ are the negative samples.

## B Additional Related Works

### B.1 Graph Structure Learning

Graph structure learning attempts to infer and optimize explicit or implicit relationships between nodes directly from data. It has delivered excellent results on graph-related tasks such as node classification, link prediction, and graph generation. Direct optimization methods including NodeFormer Wu et al. (2022), STABLE Li et al. (2022), and ProGNN Jin et al. (2020) treat the adjacency matrix as a learnable and adjust edge weights through task-driven objectives and regularization, thereby transforming graph construction from a fixed pre-processing step into a trainable module. Metric learning approaches including SUBLIME Liu et al. (2022b), GRCN Yu et al. (2020), and CoGSL Liu et al. (2022a) employ a GNN encoder to embed node features into a latent space, compute pairwise similarities, and normalize them to define connectivity. The resulting structures capture task-relevant relations while mitigating noise and structural bias. Probability estimation techniques including GEN Wang et al. (2021a), GAuG-O Zhao et al. (2021), and SGSR Zhao et al. (2023)

model links as probabilistic variables, estimate their existence likelihood via neural networks or statistical methods, and then refine the graph by selecting high-probability edges. To date, the dynamic graph structure learning methods have not been explored in ViG architectures. We therefore present the evaluation of their transferability to the vision domain.

## B.2 GRAPH CONTRASTIVE LEARNING

Graph contrastive learning uses self supervised objectives to align views of the same node, subgraph, or graph and to separate unrelated samples, producing robust and discriminative embeddings. Zhu et al. (2020) introduced node level contrastive learning with GRACE. It builds two corrupted graph views by removing edges and masking features, then maximizes agreement of node embeddings. The objective follows InfoNCE and can be viewed as maximizing mutual information. Its mix of structural and attribute changes gives diverse contexts and has links to InfoMax and triplet losses. Yu et al. (2023b) showed that contrastive training treats nodes unevenly. Hard nodes keep high InfoNCE loss under many augmentations. They define node compactness and use it as a regularizer in a novel provable training model. SOLA-GCL Peng et al. (2025) learns how to augment at the subgraph level. It detects dense subgraphs and applies learned changes such as node dropping, feature masking, and edge edits. This preserves key subgraph meaning. XSimGCL Yu et al. (2023a) found that most gains come from the contrastive loss, not from graph changes. It replaces topology edits with simple noise on embeddings and still excels. BGRL Thakoor et al. (2021) removes negatives and learns by prediction with an online and a target network. It scales well with simple changes. Collectively, these studies reveal that principled objectives and adaptive structure learning are key to advancing graph contrastive methods. Our DenseViG also contributes to the field of graph contrastive learning by decoupling EBMs into contrastive learning and generative modeling.

# C IMPLEMENTATION DETAILS

## C.1 TRAINING ALGORITHM

---

**Algorithm 1:** DenseViG training algorithm

---

**Input:** Training image $\mathcal{I}$, graph $\mathcal{G}$, GNN encoders $\phi_\theta(\cdot)$ and $\psi_\theta(\cdot)$, augmentations $t_1$, $t_2$, temperature $\tau$, relaxation $\pi$, weights $\alpha, \beta, \mu, \gamma$, batch size $N$, number of batches $B$, Langevin step size $\lambda$ and iterations $K$, and training epochs $P$

**Output:** Trained DenseViG model (updated parameters $\theta$)

Convert image $\mathcal{I}$ into graph $\mathcal{G}$, extract patch features $\mathbf{X}$, and construct the raw adjacency $\mathbf{A}$ via $k$-NN algorithm;

Randomly initialize $\phi_\theta(\cdot)$ and $\psi_\theta(\cdot)$ with hyper-parameters $\tau$, $\lambda$, $\pi$, $\alpha$, $\beta$, $\mu$, and $\gamma$;

**for** $p = 1, 2, \ldots, P$ **do**

 **for** $b = 1, 2, \ldots, B$ **do**

  Sample a mini-batch of $N$ graphs from the training examples;

  Generate two correlated graph views $\mathbf{U} = t_1(\mathcal{G}_1)$ and $\mathbf{V} = t_2(\mathcal{G}_2)$;

  Compute corresponding node representations $\mathbf{H}_1 = \varphi_\theta(\mathbf{U})$ and $\mathbf{H}_2 = \varphi_\theta(\mathbf{V})$;

  Calculate the discriminative InfoNCE loss $\mathcal{L}_{\mathrm{Disc}}(\theta)$ with Eq. 12;

  Initialize $\tilde{\mathbf{U}}^0 \sim \mathcal{N}(0, 2\lambda)$;

  **for** $k = 1, 2, \ldots, K$ **do**

   Sample $\tilde{\mathbf{U}}^k$ from relay buffer;

   Update $\tilde{\mathbf{U}}^{k+1}$ from $\tilde{\mathbf{U}}^k$ with Eq. 3;

  **end**

  Calculate the generative EBM loss $\mathcal{L}_{\mathrm{Gen}}(\theta)$ with Eq. 11;

  Calculate the complete DEL objective $\mathcal{L}_E(\theta) = \mathcal{L}_{\mathrm{Disc}}(\theta) + \alpha \mathcal{L}_{\mathrm{Gen}}(\theta)$;

  Build cosine similarity matrix $\mathbf{C}$ and draw relaxed adjacency $\mathbf{S}$ with Eq. 10;

  Symmetrize and normalize $\mathbf{S}$ to form $\mathbf{A}^*$ and compute $\mathbf{H}^* = \psi_\theta(\mathbf{X}, \mathbf{A}^*)$;

  Compute the task-specific loss $\mathcal{L}_T$ and form the global objective $\mathcal{L}_G$;

  Update model parameters $\theta$ by gradient descent to minimize $\nabla_\theta \mathcal{L}_G$;

 **end**

**end**

---

Algorithm 1 outlines training for DenseViG. Each iteration samples a mini-batch of images, converts them to graphs with patch features and a $k$-NN adjacency, and produces two augmented views. A shared GNN encoder $\phi_\theta(\cdot)$ yields node representations. We calculate the contrastive energy learning objective with a discriminative InfoNCE term in Eq. 12 and a generative term in Eq. 11. The generative negatives are synthesized from Gaussian noise via $K$ steps of Langevin dynamics from Eq. 3. We then refine the graph structure. From the cosine similarity matrix we draw a relaxed adjacency with the Gumbel Sigmoid rule in Eq. 10, symmetrize and normalize it to obtain the refined adjacency, and compute the corresponding representation with the GNN encoder $\psi_\theta(\cdot)$ for the task head. The global loss contains the task objective, the weighted DEL terms, and a sparsity regularizer on the relaxed edges. Model parameters are updated by gradient descent.

## C.2 MODEL CONFIGURATION

We scale the DenseViG to five sizes Ti, S, M, B, and L under a four stage pyramid ViG, as shown in Table 8. A DenseViG block applies a Graph encoder followed by an FFN. The encoder builds a $k$-NN graph in feature space with size 9 and performs multi head updates with 4 heads. Linear projections are applied before and after the graph convolution. The encoder expands channels and then projects back. We use GELU activations, LayerNorm, residual connections, and DropPath. The FFN has two linear layers with an expansion ratio $d_E = 4$.

Table 8: Configurations of DenseViG series. All variants use $k$-NN size 9, 4 heads, and input resolution $224 \times 224$. Stage 1 applies a convolutional stem to produce $56 \times 56$ tokens ($H/4 \times W/4$). Stages 2–4 downsample with stride-2 projections to $28 \times 28$, $14 \times 14$, and $7 \times 7$ tokens ($H/8$, $H/16$, $H/32$), widening channels at each stage. $d_D$ denotes the feature dimension, $d_F$ the representation dimension, and $d_E$ the FFN expansion dimension ratio. 'Ti' denotes tiny, 'S' small, 'M' medium, 'B' base, and 'L' large.

| Stage | Tokens | Layer specification | | DenseViG variants | | | | |
|---|---|---|---|---|---|---|---|---|
| | | | | Ti | S | M | B | L |
| 1 | $\frac{H}{4} \times \frac{W}{4}$ | Input | Dimension $d_D$ | 192 | 320 | 640 | 768 | 1024 |
| | | | Patch Size | Stem, $56 \times 56$, stride 4 | | | | |
| | | DenseViG Block | Dimension $d_F$ | 48 | 80 | 96 | 128 | 192 |
| | | | Dimension $d_E$ | 4 | | | | |
| | | | Blocks | 2 | 2 | 2 | 2 | 2 |
| 2 | $\frac{H}{8} \times \frac{W}{8}$ | Input | Patch Size | Downsample, $28 \times 28$, stride 2 | | | | |
| | | DenseViG Block | Dimension $d_F$ | 96 | 160 | 192 | 256 | 384 |
| | | | Dimension $d_E$ | 4 | | | | |
| | | | Blocks | 2 | 2 | 2 | 2 | 2 |
| 3 | $\frac{H}{16} \times \frac{W}{16}$ | Input | Patch Size | Downsample, $14 \times 14$, stride 2 | | | | |
| | | DenseViG Block | Dimension $d_F$ | 240 | 400 | 384 | 512 | 768 |
| | | | Dimension $d_E$ | 4 | | | | |
| | | | Blocks | 6 | 6 | 16 | 18 | 24 |
| 4 | $\frac{H}{32} \times \frac{W}{32}$ | Input | Patch Size | Downsample, $7 \times 7$, stride 2 | | | | |
| | | DenseViG Block | Dimension $d_F$ | 384 | 640 | 768 | 1024 | 1536 |
| | | | Dimension $d_E$ | 4 | | | | |
| | | | Blocks | 2 | 2 | 2 | 2 | 2 |
| Params (M) | | | | 11.1 | 28.5 | 53.3 | 98.5 | 120.3 |
| FLOPs (G) @ $224 \times 224$ | | | | 1.9 | 5.3 | 9.6 | 18.4 | 24.9 |

All variants have four stages with output resolutions $H/4 \times W/4$, $H/8 \times W/8$, $H/16 \times W/16$, and $H/32 \times W/32$. For input $224 \times 224$, a three layer $3 \times 3$ convolutional stem reduces the input to quarter resolution and produces $56 \times 56$ tokens, thus $N$=3136. Later stages reduce the token grid by spatial downsampling. After each stage, a stride 2 $3 \times 3$ projection reduces the number of tokens by a factor of 4 and increases the representation dimension $d_F$. The block schedule is 2 in Stage 1, 2 in Stage 2, model dependent counts in Stage 3, and 2 in Stage 4, with exact values in Table 8. We

use a linear DropPath schedule with end rates 0.1, 0.2, 0.3, 0.5, and 0.6 for Ti through L. Absolute positional encoding is used, with additional relative positional encoding within the pyramid.

## C.3 Experiment Setting

We adopt a unified training pipeline for DenseViG on classification, detection, and segmentation. Hyper-parameters follow the main ablations. Optimizer is AdamW with $\beta_1$=0.9, $\beta_2$=0.99, weight decay 0.05. We fix $\tau$=0.1, $\alpha$=1.0, $\beta$=$10^{-5}$, $\mu$=0.1, $\gamma$=$10^{-4}$, $k$=5 Langevin iterations and $\lambda$=0.05 step size for negative sampling, relaxation $\pi$=0.1 for structure refinement. We maintain an EMA of weights with decay 0.999. We apply stochastic depth with drop path rate 0.1. Table 9 summarizes the training hyper-parameters for each task. In the following, we detail any task-specific setup differences, while other settings remain consistent across tasks.

Table 9: Unified training hyper-parameter settings for DenseViG across ImageNet-1K classification, MS COCO detection, and ADE20K segmentation. Others not listed are shared across the three tasks.

| Task | Classification (ImageNet-1K) | Detection (MS COCO) | Segmentation (ADE20K) |
|---|---|---|---|
| Optimizer | AdamW ($\beta_1$=0.9, $\beta_2$=0.99) | AdamW (same as left) | AdamW (same as left) |
| Weight decay | $5 \times 10^{-2}$ | $5 \times 10^{-2}$ | $5 \times 10^{-2}$ |
| Layer-wise LR decay | — | 0.75 per layer (fine-tune) | 0.75 per layer (fine-tune) |
| Initial learning rate | $1 \times 10^{-3}$ | $2 \times 10^{-4}$ | $2 \times 10^{-4}$ |
| Training epochs | 310 | 12 ($1\times$ schedule) | 16 (fine-tune) |
| Batch size | 64 | 16 | 16 |
| Input resolution | $224 \times 224$ | $1280 \times 800$ | $512 \times 512$ |
| LR schedule | Cosine annealing | Step decay (8, 11 ep) | Step decay (proportional) |
| Data augmentation | RandAug, CutMix, RandomResizedCrop | RandomFlip, Multi-scale jittering | RandomFlip, RandomResizedCrop |
| Label smoothing | 0.1 | — | — |
| EMA decay rate | 0.999 | 0.999 | 0.999 |
| Drop-path rate | 0.1 | 0.1 | 0.1 |
| $\mathcal{L}_E$ temperature $\tau$ | 0.1 | 0.1 | 0.1 |
| $\mathcal{L}_E$ balance $\alpha$ | 1.0 | 1.0 | 1.0 |
| $\mathcal{L}_E$ balance $\beta$ | $1 \times 10^{-5}$ | $1 \times 10^{-5}$ | $1 \times 10^{-5}$ |
| $\mathcal{L}_G$ weight $\mu$ | 0.1 | 0.1 | 0.1 |
| $\mathcal{L}_G$ sparsity $\gamma$ | $1 \times 10^{-4}$ | $1 \times 10^{-4}$ | $1 \times 10^{-4}$ |
| Langevin iterations $k$ | 5 | 5 | 5 |
| Langevin step size $\lambda$ | 0.05 | 0.05 | 0.05 |
| Relaxation temp. $\pi$ | 0.1 | 0.1 | 0.1 |

**ImageNet 1K classification** We train from scratch for 310 epochs on $224 \times 224$ images. Initial learning rate is $1 \times 10^{-3}$ with a cosine schedule for total batch size 64 on 8 RTX 3090 GPUs. Augmentations include RandAugment, CutMix, RandomResizedCrop, and label smoothing 0.1. The backbone uses ViG initialization without pretraining.

**MS COCO detection** We fine tune ImageNet pretrained DenseViG with MMDetection, using RetinaNet and Mask R-CNN heads while keeping the backbone and Dense module fixed. Training runs for 12 epochs at $1280 \times 800$. AdamW and regularization match classification. We use layer wise learning rate decay of 0.75 per ViG layer from output to input. Initial learning rate is $2 \times 10^{-4}$ with step drops at epochs 8 and 11, batch size 16. Augmentations are RandomFlip and Multi-scale jittering. EMA is enabled. Losses are sigmoid focal for classification and a weighted sum of $\ell_1$ and GIoU for boxes.

**ADE20K segmentation** We fine tune ImageNet pretrained DenseViG with MMSegmentation for 16 epochs on $512 \times 512$ random crops. Optimizer, EMA, and layer wise decay match detection. Initial learning rate is $2 \times 10^{-4}$ with a proportional multi step schedule, batch size 16. Augmentations are RandomFlip and RandomResizedCrop. We attach linear MLP, PPM, Semantic FPN, and UPerNet heads. The backbone is fine tuned with the pixel wise cross entropy loss.

## D  MORE EXPERIMENT RESULTS

### D.1  OBJECT DETECTION AND INSTANCE SEGMENTATION

We evaluate object detection and instance segmentation on MS COCO using RetinaNet and Mask R-CNN, as stated in Table 10 and Table 11. All models use ImageNet 1K pretraining and follow the MMDetection recipe. Both tables report parameters, FLOPs, and COCO metrics, with RetinaNet giving $AP^{box}$ averaged over IoU 0.50 to 0.95, $AP^{box}$ at IoU 0.50 and 0.75, and $AP^{box}$ at small, medium, and large sizes, and Mask R-CNN giving both $AP^{box}$ and $AP^{mask}$ with the same breakdown.

Under the same protocol, DenseViG-S improves both tasks across heads and schedules. With Reti-naNet 1× it reaches 44.6 $AP^{box}$, exceeding DVHGNN-S by 1.3 $AP^{box}$, with consistent gains at $AP^{box}_{50}$ and $AP^{box}_{75}$ and the largest gain on large objects with plus 2.3 $AP^{box}_{L}$. The 3× with MS setting lifts this to 47.0 $AP^{box}$. With Mask R-CNN 1×, DenseViG-S attains 46.4 $AP^{box}$ and 42.7 $AP^{mask}$, im-proving over DVHGNN-S by 1.6 $AP^{box}$ and 2.5 $AP^{mask}$. The 3× with MS schedule reaches 50.2 $AP^{box}$ and 45.2 $AP^{mask}$. These gains come with similar parameters and FLOPs, which shows that DenseViG improves the detection and segmentation performance with high efficiency and scales well with stronger heads and longer training.

Table 10: Object detection on MS COCO. All backbones are ImageNet-1K pretrained. "1×" = 12 epochs; "3×" = 36 epochs; "MS" = multi-scale training. **AP**: mean over IoU 0.50:0.95 with step 0.05 and over categories. $AP^{box}$: bounding boxes. $AP^{box}$50/75: AP at IoU 0.50 and 0.75. $AP^{box}S/M/L$: COCO area; $S$ if area $< 32^2$, $M$ if $32^2 \le$ area $< 96^2$, $L$ if area $\ge 96^2$, in $px^2$.

| Backbone | Params | FLOPs | RetinaNet 1x | | | | | | RetinaNet 3x + MS | | | | | |
|---|---|---|---|---|---|---|---|---|---|---|---|---|---|---|
| | | | $AP^{box}$ | $AP^{box}_{50}$ | $AP^{box}_{75}$ | $AP^{box}_{S}$ | $AP^{box}_{M}$ | $AP^{box}_{L}$ | $AP^{box}$ | $AP^{box}_{50}$ | $AP^{box}_{75}$ | $AP^{box}_{S}$ | $AP^{box}_{M}$ | $AP^{box}_{L}$ |
| ResNet-50 | 38M | 239G | 36.3 | 55.3 | 38.6 | 19.3 | 40.0 | 48.8 | 39.0 | 58.4 | 41.8 | 22.4 | 42.2 | 51.6 |
| PVT-S | 34M | 227G | 40.4 | 61.3 | 43.0 | 25.0 | 42.9 | 55.7 | 42.2 | 62.7 | 45.0 | 26.2 | 45.2 | 57.2 |
| Swin-T | 39M | 245G | 41.5 | 62.1 | 44.2 | 25.1 | 44.9 | 55.5 | 43.9 | 64.8 | 47.1 | 28.4 | 47.2 | 57.8 |
| PyramidViG-S | 36M | 240G | 41.8 | 63.1 | 44.7 | 28.5 | 45.4 | 53.4 | 44.2 | 65.8 | 47.6 | 31.8 | 47.7 | 55.7 |
| PViHGNN-S | 38M | 244G | 42.2 | 63.8 | 45.1 | 29.3 | 45.9 | 55.7 | 44.6 | 66.5 | 48.0 | 32.6 | 48.2 | 58.0 |
| DVHGNN-S | 38M | 242G | 43.3 | 64.3 | 46.3 | 28.3 | 47.9 | 54.6 | 45.7 | 67.0 | 49.2 | 31.6 | 50.2 | 56.9 |
| DenseViG-S | 40M | 247G | **44.6** | **66.1** | **47.8** | **29.7** | **48.5** | **56.9** | **47.0** | **68.8** | **50.7** | **33.0** | **50.8** | **59.2** |

Table 11: Object detection and instance segmentation on MS COCO. All backbones are ImageNet-1K pretrained. "1×" = 12 epochs; "3×" = 36 epochs; "MS" = multi-scale training. $AP^{mask}$: instance masks. $AP^{mask}_{50/75}$: AP at IoU 0.50 and 0.75.

| Backbone | Params | FLOPs | Mask R-CNN 1x | | | | | | Mask R-CNN 3x + MS | | | | | |
|---|---|---|---|---|---|---|---|---|---|---|---|---|---|---|
| | | | $AP^{box}$ | $AP^{box}_{50}$ | $AP^{box}_{75}$ | $AP^{mask}$ | $AP^{mask}_{50}$ | $AP^{mask}_{75}$ | $AP^{box}$ | $AP^{box}_{50}$ | $AP^{box}_{75}$ | $AP^{mask}$ | $AP^{mask}_{50}$ | $AP^{mask}_{75}$ |
| ResNet-50 | 44M | 260G | 38.0 | 58.6 | 41.4 | 34.4 | 55.1 | 36.7 | 41.0 | 61.7 | 44.9 | 37.1 | 58.4 | 40.1 |
| PVT-S | 44M | 245G | 40.4 | 62.9 | 43.8 | 37.8 | 60.1 | 40.3 | 43.0 | 65.3 | 46.9 | 39.9 | 62.5 | 42.8 |
| Swin-T | 48M | 264G | 42.2 | 64.6 | 46.2 | 39.1 | 61.6 | 42.0 | 46.0 | 68.1 | 50.3 | 41.6 | 65.1 | 44.9 |
| PyramidViG-S | 46M | 259G | 42.6 | 65.2 | 46.0 | 39.4 | 62.4 | 41.6 | 46.4 | 68.8 | 50.0 | 41.9 | 65.9 | 44.4 |
| PViHGNN-S | 48M | 262G | 43.1 | 66.0 | 46.5 | 39.6 | 63.0 | 42.3 | 46.9 | 69.6 | 50.5 | 42.1 | 66.5 | 45.1 |
| DVHGNN-S | 49M | 261G | 44.8 | 66.8 | 49.0 | 40.2 | 63.5 | 43.1 | 48.6 | 70.4 | 53.0 | 42.7 | 67.0 | 45.9 |
| DenseViG-S | 51M | 266G | **46.4** | **68.7** | **50.6** | **42.7** | **65.8** | **45.4** | **50.2** | **72.3** | **54.6** | **45.2** | **69.3** | **48.2** |

