# OpenReview forum: "DenseViG: Decoupled Energy-guided Graph Structure Refinement for Vision GNNs"
_ICLR.cc/2026/Conference — ICLR 2026 Conference Withdrawn Submission_

### Official Review · Reviewer_xEoK · 2025-10-29

**Soundness:** 3
**Presentation:** 2
**Contribution:** 2
**Rating:** 4
**Confidence:** 3

**Summary:**

The paper presents DenseViG (Decoupled ENergy learning guided Structure rEfinement for improving ViG), a method for graph structure refinement within Vision Graph Neural Networks (ViGs). The work is motivated by the fact that existing ViGs rely on static strategies for graph construction (such as k-nearest neighbors), which lack learnable parameters and prevent end-to-end gradient updates for the graph structure. DenseViG addresses this by introducing a refinement module integrated into each ViG stage, which dynamically modifies the graph structure (adding or removing edges) based on node similarity. This refinement is guided by a Decoupled Energy Learning (DEL) loss function, which draws inspiration from prior work on Energy-based Contrastive Learning for GNNs (Zeng et al., AAAI’25). The DEL paradigm approximates a target distribution by splitting the optimization into two losses: (a) an Energy-Based Model (EBM) loss, a generative approach designed to encourage similar visual patches to cluster together (by minimizing their "energy"); and (b) An InfoNCE loss, a contrastive learning component. The framework is evaluated for performance across three major vision tasks: image classification (ImageNet-1K), object detection (MS COCO), and semantic segmentation (ADE20K), using a unified set of training hyperparameters.

**Strengths:**

- The proposed framework shortens iteration time and reduces memory consumption, offering practical benefits for training with large-scale images
- The method demonstrates superior performance compared to other backbones on all three vision tasks (classification, object detection and segmentation)
- The framework is plug-and-play, allowing for easy integration into various existing ViG backbones for structure refinement.

**Weaknesses:**

- The paper's precise technical novelty requires clarification relative to the work by Zeng et al. (AAAI’25) on Energy-based Contrastive Learning for GNN Structure Refinement. A detailed, explicit discussion outlining the mathematical or conceptual differences and the specific adaptations made for the ViG domain is necessary to fully establish the unique contribution of DenseViG. It would be good to also relate that to the ablations in Table 5.
- The introduction and motivation would benefit from a more cohesive narrative flow. For example, the logical connection between the limitation of static graph construction and the necessity of the Decoupled Energy Learning loss needs to be explicitly strengthened. In addition, the link between the authors' core assumptions and the subsequent technical motivation should be made clearer.

Minor: the presentation and integration of tables within the text could be refined for improved readability.

**Questions:**

- Could the authors please highlight the specific technical advancements of the DenseViG framework? In particular, how does the proposed Decoupled Energy Learning (DEL) formulation differ from the Energy-based Contrastive Learning approach in Zeng et al., and what were the critical modifications for its successful application to ViGs?
- The presentation and writing can be improved to enhance readability throughout

---

### Official Review · Reviewer_sBAo · 2025-10-31

**Soundness:** 2
**Presentation:** 3
**Contribution:** 2
**Rating:** 4
**Confidence:** 3

**Summary:**

The paper introduces DenseViG, a Decoupled Energy-Guided Graph Structure Refinement approach for Vision Graph Networks (ViG). Its central idea is Decoupled Energy Learning (DEL), which separates an energy-based model objective into discriminative and generative components, optimized jointly with truncated Langevin sampling for negative samples. The refined energy is used to guide graph structure updates through a Gumbel–Sigmoid relaxation that adds or prunes edges dynamically. Experiments on ImageNet-1K, COCO, and ADE20K demonstrate consistent gains. However, the name 'DenseViG' contradicts the sparsifying nature of the method, and the efficiency claim remains unverified.

**Strengths:**

1. The formulation of DEL is well explained which combines energy-based and contrastive objectives in a differentiable form.

2. The integration into ViG is modular and easy to implement, requiring minimal structural change.

3. Experiments show consistently modest improvement on standard benchmarks.

**Weaknesses:**

1. Novelty of the method is limited as DEL recombines existing methods (contrastive EBMs + dynamic ViG refinement) without new theoretical insight; its decoupling restates established EBM formulations.

2. The claim of 'near-linear scalability' contradicts the explicit $𝑂(𝑁^2)$ pairwise-similarity computation; no runtime, latency, or VRAM measurements are reported to substantiate it.

3. The absence of Cityscapes or other high-resolution segmentation datasets weakens the claim of scalability for dense tasks.

4. There is no quantitative or qualitative evidence showing what the refinement learns (e.g., change in graph sparsity, edge distribution).

5. Misleading naming and framing. Despite the title DenseViG, the algorithm prunes edges, enforcing sparsity; this naming inconsistency reflects conceptual confusion about the actual mechanism.

**Questions:**

1. What is the training and inference latency (ms per image) and GPU memory overhead compared to ViG baselines at $224\times224$, $384\times384$, and $512\times512$ resolutions?

2. How do the authors ensure the truncated Langevin sampler yields meaningful negatives rather than noisy updates?

3. Can adjacency visualizations or edge statistics show what structural changes DEL induces?

4. It would be interesting to see results on Cityscapes or other large-resolution dense-prediction benchmark.

5. How is the "dense" terminology justified when the refinement process reduces connectivity?

---

### Official Review · Reviewer_sEo8 · 2025-11-01

**Soundness:** 3
**Presentation:** 3
**Contribution:** 3
**Rating:** 6
**Confidence:** 4

**Summary:**

This paper proposes DenseViG, a novel framework for enhancing ViG. Its core idea is to dynamically refine graph structures by Decoupled Energy Learning, which combines Energy-Based Models and Contrastive Learning into a unified objective. Specifically, DenseViG constructs a joint distribution over paired graph views and decouples it into a generative term and a discriminative term. This framework is integrated into existing ViG architectures in a plug-and-play manner, dynamically adding or pruning edges at each stage based on learned node similarities using a Gumbel-Sigmoid relaxation strategy. Extensive experiments on ImageNet-1K, MS COCO, and ADE20K demonstrate state-of-the-art performance in image classification, object detection, and semantic segmentation.

**Strengths:**

The paper correctly identifies the limitation of static graph construction in ViG and proposes a principled solution via dynamic refinement; by leveraging graph sparsification to reduce dimensionality, the method makes EBMs tractable for large-scale vision tasks; the experiments span three major vision benchmarks and multiple tasks, with thorough ablations and comparisons against strong baselines, demonstrating consistent improvements.

**Weaknesses:**

1.While the paper claims EBM sampling is efficient due to graph sparsification, the actual cost of k-step Langevin dynamics (k=5) is only briefly mentioned (Fig. 3). A more detailed efficiency–accuracy trade-off analysis would strengthen the claim.

2.The paper states that 8 RTX 3090 GPUs were used but omits the total training time or GPU hours required for a model (e.g., DenseViG-S) on ImageNet. The experimental analysis would be strengthened by including the total training time for the models.

3.The Related Work section would be stronger if it could be more focused on works most directly relevant to energy-based graph learning and dynamic structure refinement.

4.While GreedyViG and MobileViG are included, the paper does not deeply analyze why DenseViG outperforms them, e.g., is it the DEL objective, the relaxation strategy, or both?

**Questions:**

What is the core advantage of DenseViG: the DEL objective function itself, or the specific implementation details like the Gumbel relaxation?

---

### Note · Authors · 2025-11-28

I have read and agree with the venue's withdrawal policy on behalf of myself and my co-authors.